# A Novel Fault Diagnosis Method of Rolling Bearing Based on Integrated Vision Transformer Model

**DOI:** 10.3390/s22103878

**Published:** 2022-05-20

**Authors:** Xinyu Tang, Zengbing Xu, Zhigang Wang

**Affiliations:** 1Key Laboratory of Metallurgical Equipment and Control Technology, Wuhan University of Science and Technology, Ministry of Education, Wuhan 430081, China; tangxinyu@wust.edu.cn (X.T.); wzhigang@wust.edu.cn (Z.W.); 2Hubei Key Laboratory of Mechanical Transmission and Manufacturing Engineering, Wuhan University of Science and Technology, Wuhan 430081, China; 3The State Key Laboratory of Digital Manufacturing Equipment & Technology, Huazhong University of Science and Technology, Wuhan 430074, China

**Keywords:** vision transformer, integrated vision transformer, fault diagnosis, rolling bearing

## Abstract

In order to improve the diagnosis accuracy and generalization of bearing faults, an integrated vision transformer (ViT) model based on wavelet transform and the soft voting method is proposed in this paper. Firstly, the discrete wavelet transform (DWT) was utilized to decompose the vibration signal into the subsignals in the different frequency bands, and then these different subsignals were transformed into a time–frequency representation (TFR) map by the continuous wavelet transform (CWT) method. Secondly, the TFR maps were input with respective to the multiple individual ViT models for preliminary diagnosis analysis. Finally, the final diagnosis decision was obtained by using the soft voting method to fuse all the preliminary diagnosis results. Through multifaceted diagnosis tests of rolling bearings on different datasets, the diagnosis results demonstrate that the proposed integrated ViT model based on the soft voting method can diagnose the different fault categories and fault severities of bearings accurately, and has a higher diagnostic accuracy and generalization ability by comparison analysis with integrated CNN and individual ViT.

## 1. Introduction

The rolling bearing plays an important role in rotating machinery, its health is related directly to the overall operating conditions and the quality of the mechanical equipment. Bearing failure can lead to equipment failure and cause serious economic losses or casualties to the enterprise. Therefore, it is very important to monitor and diagnose the health status of rolling bearings through their vibration data to ensure normal production in enterprises [1,2,3,4,5,6,7].

In recent years, more and more deep learning models have been developed and applied to the fault diagnosis of rolling bearings because of the end-to-end diagnosis ability. The typical CNN neural network is widely applied in the field of bearing-fault diagnosis because of its simplistic network structure and high accuracy diagnosis ability [8,9,10,11]. Although the feature extraction capabilities of the CNN model from the one-dimensional time-series signal or two-dimensional image can be increased by continuously stacking more convolutional layers, the CNN cannot capture long-range feature information [12,13]. To solve this problem, the position embedding method is introduced into the CNN to analyze the sequential nature of the time series signal or two-dimensional image, the multihead self-attention and parallel training mechanisms are all incorporated into the CNN model to capture the globally sensitive features from the time series signal or two-dimensional image quickly [14], thus the transformer model is proposed and applied in the field of natural language processing and image recognition [15]. After that, Yifei Ding et al. [16] applied the transformer method to the field of fault diagnosis of mechanical equipment. However, the transformer has low computational efficiency and large memory consumption. In order to solve this problem, a vision transformer (ViT) which removes the decoder block of the transformer model is proposed for application in vision processing with a higher recognition performance, because it can not only inherit the multiheaded self-attention mechanism and relative position embedding method of the transformer but can also adopt the parallel learning mechanism and be prone to capture the global spatiotemporal information of an image [17,18,19]. Based on the advantages of ViT, a one-dimensional ViT architecture with multiscale convolution fusion is proposed to capture the fault features in multiple time scales with the transformer and achieve high diagnosis accuracy on the bearing fault dataset [20]. However, in the diagnosis process, ViT cannot thoroughly reveal the fault features concealed in the vibration signals, especially when fluctuations in the working conditions occur. This can affect the diagnosis performance of ViT. In addition, the diagnosis accuracy and generalization of the ViT model can be degraded because of over-fitting.

To capture more fault-related information, some time–frequency signal-processing methods, such as wavelet transform (WT), empirical mode decomposition (EMD), etc., can not only denoise the original vibration signal but also decompose the signal into different scale components which are combined with a deep learning model to extract fault features for improving the diagnosis accuracy [21,22]. However, the EMD and its variants suffer from mode mixing which decreases the decomposition performance. WT can overcome the problem, and discrete wavelet transform (DWT) can decompose the original vibration signal into the required scale components without reducing the amplitude, Continuous wavelet transform (CWT) can detect the singularity of the different scale components. Thus, the DWT over CWT can be utilized to detect the singularity of the required scale components for bearing fault diagnosis.

Integration learning has been widely applied to the fault diagnosis of bearings by flexibly fusing the preliminary diagnosis results of multiple base classifiers to obtain diagnosis results with higher accuracy and generalization ability because of the complementary classification behavior among different base classifiers. When integrated deep learning models combined with the different scale components of the original signal are utilized to diagnose the fault classes, higher diagnosis accuracy and generalization can be produced. Currently, some integrated deep learning models have been developed to apply to the field of fault diagnosis, and achieve good diagnosis results [23,24,25].

To the best of our knowledge, integrated learning has not been introduced into the ViT model to diagnose bearing faults. In order to improve the diagnosis accuracy and generalization of ViT, an integrated ViT model combined with wavelet transform and the soft voting method is proposed in this paper. The main contributions of the proposed diagnosis method are summarized as follows:(1)The integrated ViT based on the soft voting fusion method is suggested to diagnose the bearing fault with high accuracy and generalization;(2)DWT is used to decompose the original signal into different subsignals in different frequency bands and denoise the subsignals. After that, CWT is utilized to transform the subsignals into time–frequency representation (TFR) maps which can describe the singularity of the different subsignals;(3)The ViT model can dig out more hidden fault-related information from the different TFR maps of the subsignals in different frequency bands.

The rest of the paper is organized as follows: Section 2 introduces the integrated ViT model which is combined with wavelet transform and the soft voting method; Section 3 presents the fault diagnosis flowchart of the integrated ViT; Section 4 gives the fault diagnosis experimental analysis of bearings based on the integrated ViT; lastly, the Conclusion is shown in Section 5.

## 2. Integrated Vision Transformer Model

The integrated ViT model not only inherits the advantages of integrated learning but also inherits the advantages of the ViT model, which can improve the diagnosis accuracy and generalization. Figure 1 shows the proposed fault diagnosis scheme diagram of the proposed integrated ViT model. Firstly, the vibration signal is gradually truncated through the sliding time window and divided into different data segmentations (data samples) which are decomposed into *n* subsignals in different frequency bands by DWT, and these subsignals are transformed into the corresponding *n* TFR maps by CWT, and then *n* individual ViT models which consist of a linear projection of flattened patches (embedding layer), the transformer encoder and MLP head are utilized to diagnose these *n* TFR maps of subsignals to obtain the preliminary diagnosis results, respectively. Finally, the final diagnosis decision can be obtained by the soft voting method used to fuse all the preliminary diagnosis results.

### 2.1. DWT-Based Signal Decomposition

Referring to Figure 1, TFR maps of multiple subsignals in different frequency bands are input into multiple individual ViT models to diagnose the fault preliminarily. In order to reduce the influence of noise, DWT is used to decompose the original signal into different subsignals in different frequency bands without reducing the amplitude.

The discrete wavelet transform (DWT) can map any stationary or non-stationary signal to a set of base functions formed by wavelet scaling to obtain subsignals distributed in different frequency bands with complete information in the pass frequency range [26]. Based on the rules, the fault-related information in different frequency bands can be mined to diagnose the fault. Thus, through the scaling and translation of wavelet function basis and scale function, the original signal can be decomposed into different subsignals with different scales. The detailed algorithm of DWT is described as follows.

(1) Given a time series signal $a_{0}$ of length N is decomposed by the Mallat tower wavelet decomposition algorithm [27], the decomposition process can be expressed as
(1)$$\{\begin{array}{c} a_{i+1}^{\prime }=Ha_{i}\\ d_{i+1}^{\prime }=Ga_{i} \end{array}$$
where *H* denotes the low-pass filter, *G* denotes the high-pass filter, $a_{i}$ is the signal to be decomposed. $a_{i+1}^{\prime }$ and $d_{i+1}^{\prime }$ are the low-frequency and high-frequency coefficients obtained by the one-half down-sampling method, respectively.

(2) The obtained low-frequency coefficients can be decomposed repeatedly according to Equation (1). Thus, the low-frequency coefficients obtained from the decomposition of level *j* and the high-frequency coefficients obtained from the decomposition of each level are reconstructed to obtain subsignals in different frequency bands. For example, Figure 2 shows the 4-level decomposition result of the signal, *A*_4_ and a set of subsignals *D*_4_, *D*_3_, *D*_2_, *D*_1_ represent approximate signal and detailed signals with frequency from low to high respectively. The relationship between the decomposed components (subsignals) and the original signal *X* can be expressed as
(2)$$X=A_{4}+D_{4}+D_{3}+D_{2}+D_{1}$$

### 2.2. Time–Frequency Analysis Based on CWT

The time–frequency analysis method is mainly used to reveal the time–frequency representation (TFR) of the subsignals in different frequency bands which can describe the relationship between the time and the frequency. At present, there are many time–frequency analysis methods used to analyze the time–frequency characteristics of vibration signals, such as short-time Fourier transform (STFT), Wigner–Ville distribution (WVD), and continuous wavelet transform (CWT). However, the STFT is unable to locate the time and frequency of non-stationary signals accurately [28]; the WVD is prone to frequency aliasing and cross-term interference [29]. In contrast, the CWT not only has good time–frequency resolution and time–frequency localization ability, but also detects the singularity of the signal. Thus, the corresponding time–frequency map of the subsignals in different frequency bands can depict the distinguished fault-related information [30,31]. The analysis process of CWT is described as follows.

Assume that the mother wavelet or basic wavelet function *ψ* is satisfied with $\psi \in L^{1}\left(R\right)\cap L^{2}\left(R\right)$ and $\hat{\psi }\left(0\right)\in 0$, the wavelet function family can be obtained by scaling and translation of function *ψ*. The wavelet function is written as follows
(3)$$\psi_{a,b}\left(t\right)={\left|a\right|}^{-1/2}\psi \left(\frac{t-b}{a}\right),a,b\in R,a\neq 0$$
where $\left\{\psi_{a,b}\right\}$ is the analytic wavelet or continuous wavelet, *a* is the scaling factor of changing the wavelet shape, *b* is the translation factor of the wavelet shift. Thus, the CWT for an arbitrary function $f\left(t\right)\in L^{2}\left(R\right)$ can be expressed as
(4)$$W_{f}\left(a,b\right)=\langle f,\psi_{a,b}\rangle ={\left|a\right|}^{-1/2}\int_{R}f\left(t\right)\bar{\psi \left(\frac{t-b}{a}\right)}dt$$
where $\bar{\psi \left(t\right)}$ is the complex conjugate of ψ(t), the symbol $\langle f,\psi_{a,b}\rangle$ is the inner product of the function *f* and *ψ*. *W_f_* (*a*,*b*) denotes the coefficients of the wavelet function with scale *a* and offset *b*, which represent the similarity between the wavelet function and the original signal, and both *a* and *b* are continuous variables.

In order to obtain the fault-related TFR of a subsignal in different frequency bands in this paper, the CWT is used to transform the signal into a TFR, which is described as follows:

Assuming that *f*_s_ is the sampling frequency and *F_c_* is the wavelet center frequency, the actual frequency *F_a_* corresponding to scale *a* is written as
(5)$$F_{a}=F_{c}\times f_{s}/a$$

In order to make the transformed frequency sequence an equal difference sequence, the scale sequence must take the following values.
(6)c/totalscal,⋯,c/(totalscal−1),c/4,c/2,c
where *totalscal* is the length of the scale series used in the wavelet transform of the signal, which is set as 256 here, and c is a constant.

On the basis of the sampling theorem, the actual frequency corresponding to the scale *c/totalscal* should be *f_s_*/2. The value of the constant c can be calculated according to Equation (5), which can be obtained by the following equation
(7)$$c=2\times F_{c}\times totalscal$$

Accordingly, the required scale sequence is obtained by substituting Equation (7) into Equation (6).

After determining the wavelet basis function and scale, the wavelet coefficients *W_f_* (*a*,*b*) are obtained by applying the continuous wavelet transform principle of Equation (4). Then the scale sequence is converted into the actual frequency sequence *f* by Equation (5). Finally, the TFR map can be plotted.

### 2.3. Vision Transformer Model (ViT)

A transformer is a typical neural network model that relies entirely on a self-attention mechanism to establish the relationship between input and output, which can consider the global information comprehensively and be trained in parallel because of the parallel architecture that is completely different from the sequential structure of the traditional recurrent neural network (RNN). Figure 3 shows the architecture of the transformer model which mainly consists of a positional embedding layer, an encoder and a decoder. The positional embedding is used to add the relative positional information of the input data to the data processed by the embedding layer, thus, the transformer can better solve the long-time dependency problem. Based on these characteristics, the transformer can achieve good performance on much vision detection, but it requires a good deal of memory and computational power.

In order to solve this problem, the vision transformer (ViT) was proposed by Dosovitskiy [17]. ViT has been applied widely to the field of image and vision recognition because of lower computational power and memory consumption, fewer training parameters and fewer training samples. Figure 4 shows the structure of the ViT model which consists of a linear projection of flattened patches (embedding layer), a transformer encoder and an MLP head. The model’s first step is to divide an input image into a sequence of image patches. These image patches are then passed through a trained linear projection layer which plays the role of an embedding layer and outputs the vectors of fixed size. Position embeddings are linearly added to the sequence of image patches so that the images can retain their positional information. Then this new sequence of image patches is fed into the transformer encoder which is mainly composed of a multihead attention layer and a multilayer perceptron (MLP) layer; the multihead attention layer splits the inputs into several heads so that each head can learn different levels of self-attention. The outputs of all the heads are then concatenated and passed through the MLP head which is added to the transformer encoder to give the network’s output classes.

#### 2.3.1. Embedding Layer

The embedding layer is mainly for implementing the linear projection of flattened image patches and retaining the positional information and one-dimensional feature vector and class labels of the image patches. Suppose the input image $x\in R^{h\times w\times c}$, where *h* denotes the height of the image, *w* denotes the width of the image, and *c* denotes the number of channels of the image; the image is split into *N* image patches with length *p* and width *p* firstly, and then the image is flattened into a one-dimensional sequence $x_{p}\in R^{N\times \left(p*p*c\right)}$. After that, a linear projection is conducted on the one-dimensional sequence $x_{p}^{\prime }\in R^{N\times D}$, these image patches are mapped into the *D* dimension vector space.

Additionally, the class label and the positional information of the image patch are all added to the outputs of embedding layer. Thus, a new sequence of image patches that contains the image features and the positional and class label information is obtained, which is the input of the transformer encoder.

#### 2.3.2. Transformer Encoder

Each transformer encoder layer consists of multiple identical modular layers arranged in a stack, its internal structure is shown in Figure 5a. Each module layer contains two sublayers which are a multiheaded self-attention layer and MLP feed-forward network respectively, the structure of the MLP feed-forward network can be seen in Figure 5b. To improve the accuracy of the network model by increasing the depth of the network generally, each sublayer is internally connected using residuals, and layer normalization is used at the end of each sublayer to improve the training speed and generalization performance of the neural network. The output of each sublayer can be expressed as
(8)o=LayerNorm(x+Sublayer(x))
where *Sublayer*(*x*) indicates the multiheaded self-attentive function and MLP function in each sublayer, *LayerNorm*(▪) denotes the normalization function.

The internal structure of the MLP layer and the multihead self-attention layer are described as follows.

MLP layer

The internal structure of the MLP is shown in Figure 5b, which comprises a fully connected layer, GELU function, and dropout function. To improve the convergence of the network, in the feedforward layer the ViT uses the Gaussian error linear unit (GELU) activation function instead of the ReLU activation used in transformer. The output of the GELU activation is expressed as follows
(9)$$GeLU\left(x\right)=x\cdot \frac{1}{2}\left[1+erf\left(x/\sqrt{2}\right)\right]$$
where *x* denotes the input, and *erf*(▪) denotes the Gaussian error function.

Multiheaded self-attention layer

The self-attention mechanism allows the network model to extract local valid features, but a single attention mechanism can only learn relevant information in one representation space. To synthetically extract long-distance features from a global image, a multiheaded self-attention mechanism is used to jointly focus on features from different representation subspaces at different locations. The structure of the multihead attention layer is shown in Figure 6. The self-attention mechanism uses scaled dot-product attention to calculate the attention value of the feature matrix. The calculation formula of the scaled dot-product attention is written as follows [32],
(10)$$Attention\left(Q,K,V\right)=softmax\left(\frac{Q\cdot K^{T}}{\sqrt{d_{k}}}\right)\cdot V$$
(11)$$Q=X_{f}W^{Q}$$
(12)$$K=X_{f}W^{K}$$
(13)$$V=X_{f}W^{V}$$
where *Q* is the query matrix, *K* is the key matrix, *V* is the value matrix. These three matrices are obtained by multiplying the input feature matrix *X_f_* with the parameter matrices *W^Q^*, *W^K^* and *W^V^* respectively, *d* is the dimension of *Q*, *K* and *V*.

The multihead self-attention mechanism is a combination of multiple self-attention mechanisms, which use multiple self-attention heads to learn features from different representation subspaces, respectively, and then the multiple attention value is combined and transformed linearly, thus the final attention value is obtained to realize the representation under these different constraint conditions. The multihead self-attention mechanism equation can be expressed as follows
(14)$$MultiHead\left(Q,K,V\right)=Concat\left(head_{1},\cdots ,head_{h}\right)\cdot W^{O}$$
(15)$$head_{i}=Attention\left(Q\cdot W_{i}^{Q}\cdot K\cdot W_{i}^{K},V\cdot W_{i}^{V}\right)$$
where $W_{i}^{Q}$, $W_{i}^{K},$ $W_{i}^{V}$ are the weight matrices of the *i*th attention head *Q*, *K* and *V*, *W^O^* is the weight matrix of multihead attention, *h* is the number of attention heads, *Concat* function is to concatenate the output values of each attention head.

#### 2.3.3. MLP Head

Generally, the standard MLP head layer of ViT consists of a fully connected layer and activation function, which is used to diagnose the fault classes. In order to reduce the calculation workload of the ViT model, a Gaussian error linear unit (GELU) activation function is adopted in this paper. Thus, the input data processed by the transformer encoder layer is input into the MLP head to obtain the probability value of each fault class, and the final fault class can be obtained according to the maximum probability value.

### 2.4. Decision Fusion Based on Soft Voting Method

When the output of a single classifier is the probability value of each fault class, the fusion is most generally performed by the soft voting method [33]. Considering the classification output of ViT is the probability value corresponding to each fault class, the soft voting method is adopted to fuse all the outputs of multiple ViTs to obtain the final diagnosis results. The fusion process based on the soft voting method is shown in Figure 7. Suppose the output probability vectors *y_k_*(*x^k^*) of the time–frequency map *x^k^* produced by the *k*th base classifier {*M*^(*k*)^}, the maximum value *Y*(*X*) is taken as the final classification result, which is defined as follows
(16)$$Y\left(X\right)=max\left\{\frac{1}{K}\overset{K}{\underset{k-1}{\sum }}y_{k}\left(x^{k}\right)\right\}\;$$
where *x^k^* denotes the CWT time–frequency map of the subsignal in the *k*th frequency bands decomposed by DWT on the original data samples *x*, *max*() denotes the maximum function, *K* is the number of base classifiers.

## 3. Diagnosis Method Based on Integrated ViT Model

The diagnosis flowchart based on the proposed integrated ViT model is shown in Figure 8. The collected vibration signal is segmented into different data samples by the sliding time window, and then these data samples are divided into a training dataset and test dataset. By the DWT and CWT method, the data samples in the training dataset are decomposed into different subsignals in different frequency bands to obtain the different time–frequency representation (TFR) maps which are input into individual ViT models respectively, and then the multiple trained ViT models can be obtained. In the same way, the TFR maps of different subsignals in different frequency bands in the test dataset are also obtained, which are input to the multiple trained ViT models, respectively, to obtain the preliminary diagnosis results. After that, the final diagnosis result is obtained by using the soft voting method to fuse all the preliminary diagnosis results.

In addition, the number of the individual ViT model is determined by the number of the subsignals in different frequency bands, which is set as five here. The loss functions of all ViT models are all cross entropy loss functions which are written as follows:(17)$$Loss=-\frac{1}{N}\underset{i}{\sum }L_{i}=-\frac{1}{N}\overset{N-1}{\underset{i=0}{\sum }}\overset{K-1}{\underset{k=0}{\sum }}y_{i,k}\cdot log\left(p_{i,k}\right)\;$$
where *K* is the number of fault classes, *N* is the number of training samples, *y_i_*_,*k*_ is the symbolic function (0 or 1), which takes 1 if the true fault class of sample *i* is equal to *k*, take 1, otherwise it is 0. *p_i_*_,*k*_ denotes the probability value of fault class *k* that the data sample *i* belongs to. The parameters of each individual ViT model are trained by the TFR maps, respectively.

## 4. Fault Diagnosis of Rolling Bearing

### 4.1. Acquisition of Bearing Vibration Signal

In order to verify the effectiveness of the proposed integrated ViT model for the fault diagnosis of the rolling bearing, the diagnosis method was utilized to diagnose the fault signals obtained from the Case Western Reserve University (CWRU) Bearing Data Center [34,35]. As shown in Figure 9, the experimental equipment comprised a motor, rolling bearing, torque sensor, and dynamometer. The bearing under test was 6205-2RS JEM KSF, a deep groove ball bearing. The drive end (DE) bearing fault signal with a sampling frequency 48 kHz, a spindle speed 1797 r/min and a load 0 hp was collected.

The electrical discharge machining (EDM) method was used to simulate the different fault categories and severity of the bearings. Three different diameters of single point damage (0.18 mm, 0.36 mm and 0.53 mm) on the inner ring, outer ring and rolling element of the bearing were introduced, respectively. The statistics of the dataset are described in Table 1. The dataset contains 10 fault classes, the number of data samples for each fault class was 500, of which 350 were training samples and 150 were test samples. Thus, the total number of training data samples and test samples were 3500 and 1500, respectively. In addition, the number of data points per data sample was 1024.

### 4.2. Wavelet Transform Analysis of Vibration Signal

#### 4.2.1. Obtaining Subsignals in Different Frequency Bands Based on DWT

Considering that the shape of Daubechies (Db) wavelet function is similar to the waveform of bearing vibration signal, and in order to obtain a better frequency band division effect and reduce the calculation time, DWT based on Db5 wavelet function was used to decompose the data sample into different subsignals in frequency bands in this paper. Figure 10 shows the four level decomposition results of the bearing vibration signal which are the original signal, detail signals D1, D2, D3, D4 and approximate signal A4, respectively. It can be seen that the different detail subsignals and approximate subsignals depict the vibration characteristics from different scales such as the vibration amplitudes and frequency, but the relationship between time and frequency cannot be shown.

#### 4.2.2. Time–Frequency Analysis Based on CWT

In order to obtain the TFR maps of subsignals in different frequency bands to describe the relationship between time and frequency, the different detail subsignals D1, D2, D3, D4 and the approximation subsignal A4 was transformed by the CWT based on the cmor3-3 wavelet basis function which was selected because its shape is similar to the impact signal of a bearing fault. Figure 11 shows the TFR maps of the original vibration signal and different approximate and detail subsignals. From the figure, it can be seen that the frequency components of the original vibration signal contained the frequency components of approximate subsignals A4 and detail subsignals D1 and D2, but the frequency components of detail subsignals D1 and D2 were obviously different from the frequency components of the original vibration signal, namely that these detail and approximate subsignals could reveal more frequency information on the bearing, and they could depict the relationship between time and frequency. All these demonstrated that the CWT method can dig out more fault-related information from the original vibration signal.

### 4.3. Diagnosis Analysis

In order to verify the effectiveness of the integrated ViT models based on wavelet transform and the soft voting method, the TFRs of five different detail and approximation subsignals in different frequency bands of size 64 × 64 × 3, were input to five individual ViT models, respectively, to preliminarily diagnose the fault of the bearing, and then the soft voting method was used to fuse all the preliminary diagnosis results to obtain the final diagnosis result.

Figure 12 shows the diagnosis results using the individual ViT model with different detail and approximation subsignals, respectively, decomposed from some vibration data samples, where the *X*-axis denotes the test data sample number and *Y*-axis denotes the fault-class labels. From the figure, it can be seen that the diagnosis accuracy produced by ViT model with the detail subsignal D1 achieved 95.07%, which is the highest among all diagnosis accuracy produced by all the individual ViT models with other detailed and approximate subsignals; the diagnosis accuracy produced by the individual ViT model with the detailed subsignal D2, D3 and D4 was 94.80%, 76.47%, respectively, the diagnosis accuracy produced by the individual ViT model with approximate subsignal A4 was 74.40%, which was only higher than that of the individual ViT model with D4, the diagnosis accuracy of the individual ViT model with D4 was the lowest, only 73.73%. This is mainly because the different detailed and approximate subsignals in the different frequency bands contained different amounts of fault-related information, and the amount of fault information in different frequency bands can affect the diagnosis accuracy of the individual ViT model directly.

The final diagnosis results of the integrated ViT model using the soft voting method for decision making to fuse all the preliminary results of five individual ViT models with five different detailed and approximate subsignals in different frequency bands were obtained and are shown in Figure 13. In addition, to further validate the effect of integrated ViT, the diagnosis results of the integrated ViTs with the different numbers of individual ViT models are also shown in Figure 13, where the *X*-axis and the *Y*-axis indicate the names of the detailed and approximate subsignals and the diagnostic accuracy, respectively; the histogram indicates the diagnosis accuracy of the individual ViT with different TFR maps of the subsignals in different frequency bands, the curve indicates the diagnostic accuracy of the integrated ViT model with the first n TFR maps of subsignals in different frequency bands. From the figure, it can be seen that the diagnosis accuracy of the integrated ViT models increased with the number of individual ViT models involved in the integration. The diagnosis accuracy of the integrated ViT models with first two individual ViT models can reach 99.13%, and the diagnosis accuracy of integrated ViT with the five individual ViT models achieved 100.00%, which exceeded the accuracy of all individual ViT models and the accuracy of other integrated ViT models with different numbers of individual ViT models. The diagnostic accuracy of all the integrated ViT models with different numbers of individual ViT models was consistently higher than the highest diagnostic accuracy of the individual ViT model. All these indicate that the integrated ViT model based on the soft voting method has a superior diagnosis performance to the individual ViT model, integrated learning can improve the diagnosis accuracy of the individual ViT model.

#### 4.3.1. Comparison with Other Integrated Models and Individual Models

To verify the superiority of the integrated ViT model, the TFRs of the subsignal in different frequency bands were also input into the integrated CNN models based on the soft voting method to diagnose the bearing fault. The structure parameters of the individual CNN are shown in Table 2.

Figure 14 shows the diagnosis accuracy of the integrated CNN models using the different numbers of individual CNN models with the TFRs of subsignals in different frequency bands; the histogram indicates the diagnosis accuracy of the individual CNN with different TFR maps of subsignals in different frequency bands, the curve indicates the diagnosis accuracy of the integrated CNN models using the first n individual CNN models with the TFR of different subsignals. It can be seen that the diagnosis accuracy of the integrated CNN models increased with the number of individual CNN models involved in the integration. The highest diagnosis accuracy obtained by the integrated CNN model using the different numbers of individual CNN models was 99.13%, which was higher than the highest diagnosis accuracy of the individual CNN model, i.e., 96.20%. From Figure 12 and Figure 13, it can be seen that the diagnosis accuracy of the integrated CNN model was always lower than that of the integrated ViT model with the same number of individual diagnosis models, and the highest diagnosis accuracy of integrated CNN model was lower than the highest diagnosis accuracy of the integrated ViT model. This could indicate that the integrated ViT has superior diagnosis ability.

In addition, to verify the stability of the proposed integrated ViT model, the five diagnosis tests were conducted by the individual ViT model, the integrated ViT model and the integrated CNN model, respectively. Table 3 shows the average diagnostic accuracy, the minimum and maximum accuracy produced by the three diagnosis models. From the table, it can be seen that the mean, minimum and maximum values of the diagnosis accuracy of the integrated ViT model were 99.87, 99.47 and 100.00%, respectively, which were all the highest among the corresponding accuracies of three diagnosis models, and the mean, minimum and maximum values of diagnosis accuracy of the integrated CNN model were higher than those of the individual ViT model, respectively. All these demonstrate that the integrated ViT model has higher diagnosis accuracy and diagnosis stability compared with the integrated CNN and individual ViT, and integrated learning can further improve the diagnosis accuracy and stability of the individual ViT.

#### 4.3.2. Generalization Analysis of the Integrated ViT

To validate the generalization of the proposed integrated ViT model based on the soft voting method, the fault diagnosis analysis of the three diagnosis models was conducted on three different datasets under three different working conditions (0 hp load and 1797 rpm, 1 HP load and 1772 rpm and 2 HP load and 1750 rpm) which are referred to as dataset 1, dataset 2 and dataset 3, for convenience. Each dataset contained 10 fault classes, the number of samples for each fault class was 200, including 140 training samples and 60 test samples. Thus, each dataset had 1400 training samples and 600 test samples. Each sample had 1024 data points. Table 4 shows the diagnosis results of the integrated ViT, integrated CNN and individual ViT on the three different datasets.

From Table 4, it can be seen that the diagnosis accuracy of the integrated ViT on the three datasets was 100.00, 99.67 and 99.83%, respectively; the maximum difference among these three diagnosis accuracies is 0.33%, and the minimum difference among them is 0.17%. The diagnosis accuracy of the integrated CNN on the three datasets was 99.17, 99.33 and 98.33%, respectively; the maximum difference among these three average diagnosis accuracies is 1%, and the minimum difference among them is 0.16%. The diagnosis accuracy of the individual ViT on the three datasets was 98.83, 98.67 and 97.83%, the maximum difference among these three diagnoses accuracies is 1%, and the minimum difference among them is 0.16%. The diagnosis accuracies of the integrated ViT on the three datasets are the highest among the three diagnosis models respectively, the diagnosis accuracies of the individual ViT on the three datasets are the lowest among the three diagnosis models respectively. In addition, the maximum difference of the diagnosis accuracy of the integrated ViT on the three datasets was the lowest among the three diagnosis models, the minimum difference of the diagnosis accuracy of the integrated ViT was only 0.01% higher than that of integrated CNN and individual ViT, respectively. All these can demonstrate that the integrated ViT has stronger diagnosis generalization than the integrated CNN and individual ViT, and furthermore, has the highest diagnosis accuracy among the three methods.

## 5. Conclusions

This paper proposes an integrated ViT model with the TFR maps of subsignals in different frequency bands based on the soft voting method to diagnose bearings. In the diagnosis process, DWT is used to decompose the vibration signal into different subsignals in different frequency bands and CWT is utilized to obtain TFRs of subsignals in different frequency bands, and then the TFR maps of the different subsignals are input into multiple individual ViT models to diagnose the fault preliminarily, and lastly, the final diagnosis result is obtained by the fusion method based on the soft voting method.

The effectiveness and generalization of the proposed integrated ViT model were verified by comparison with the integrated CNN model based on the soft voting method and individual ViT model. Through a multifaceted comparison of the three methods on different experimental datasets, the diagnosis results demonstrated that the proposed integrated ViT has higher diagnosis accuracy and generalization than the integrated CNN and individual CNN model for fault diagnosis of rolling bearings. All these show that the integrated ViT has a promising development prospect in the field of fault diagnosis of mechanical equipment. However, it was found that the number of ViT models used for integrated learning affected the diagnosis accuracy in the process of fault diagnosis, so how to select the number of ViT models with subsignals in different frequency bands will be studied further in future.

## Figures and Tables

**Figure 1 sensors-22-03878-f001:**
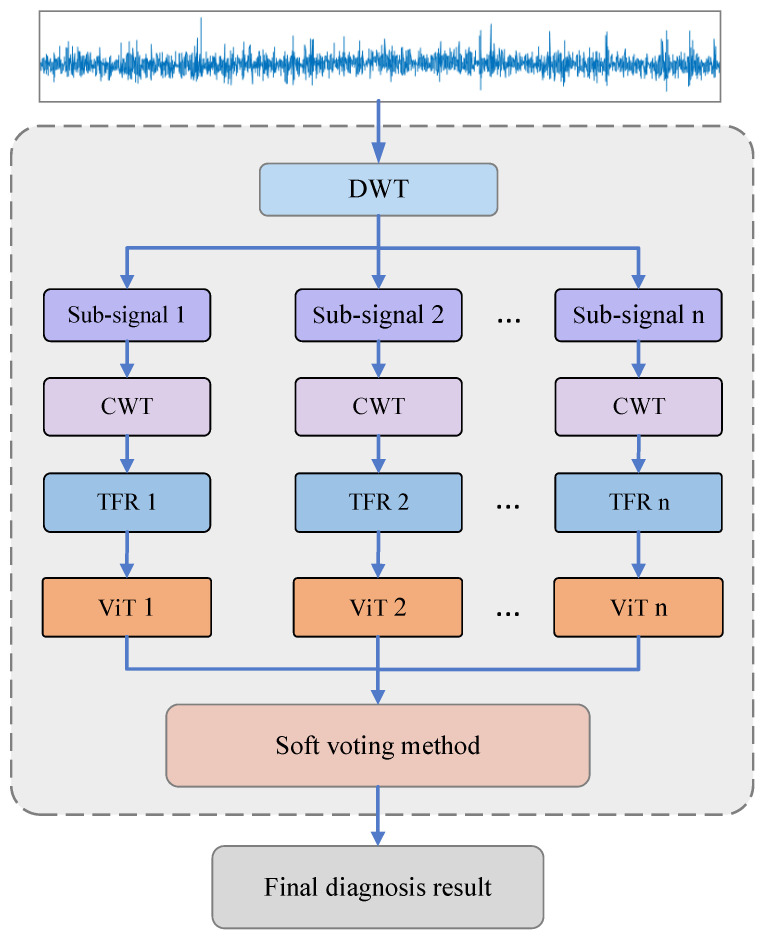
The diagnosis scheme diagram of the proposed integrated ViT.

**Figure 2 sensors-22-03878-f002:**
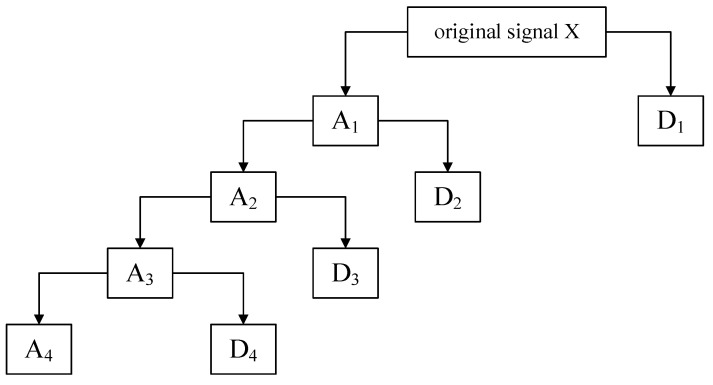
The DWT decomposition schematic. *A_i_* indicates the *i*th layer approximate signal, *D_i_* indicates the *i*th layer detailed signal.

**Figure 3 sensors-22-03878-f003:**
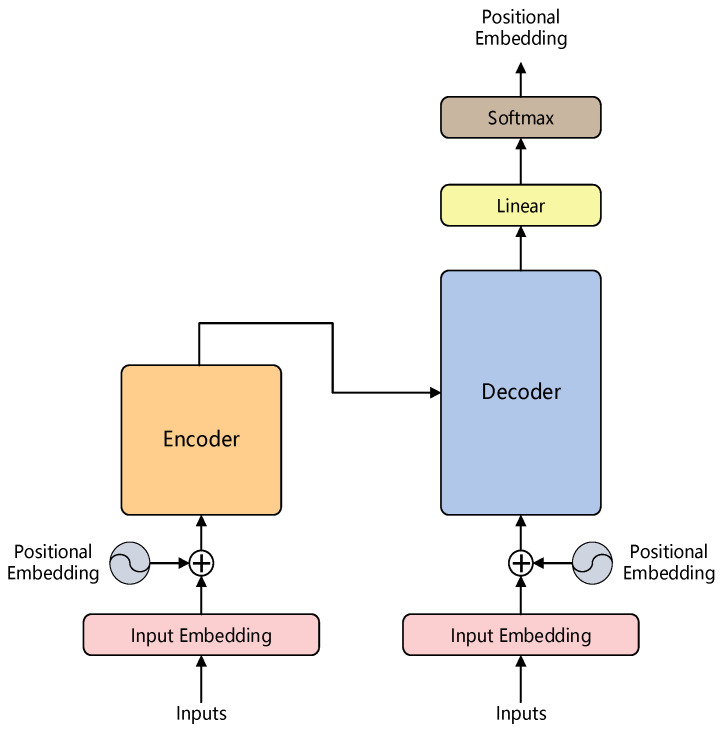
The architecture of transformer model.

**Figure 4 sensors-22-03878-f004:**
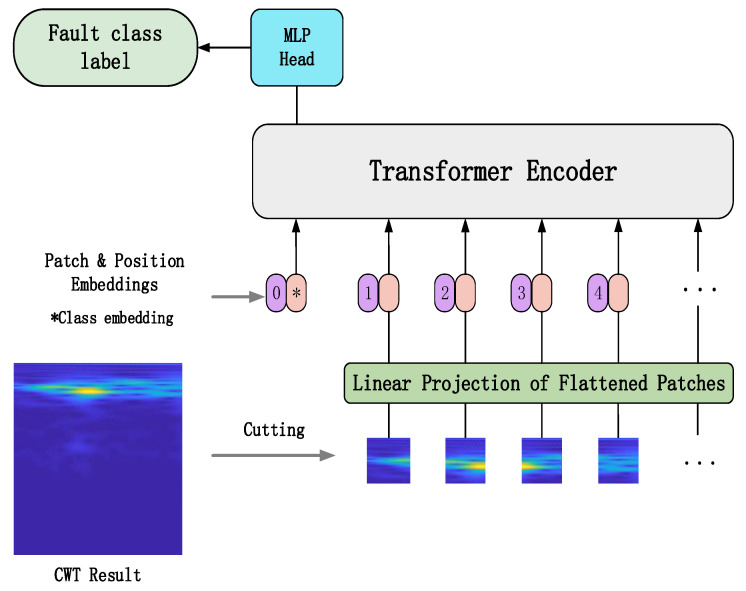
The structure of ViT model. * denotes the embedded class label vector.

**Figure 5 sensors-22-03878-f005:**
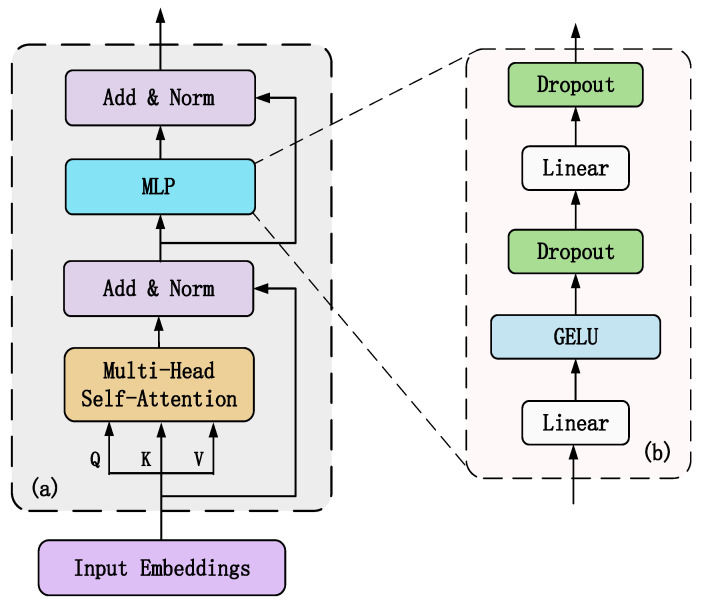
The architecture of the transformer encoder module. (**a**) the architecture of the transformer encoder module, (**b**) the internal architecture of the MLP.

**Figure 6 sensors-22-03878-f006:**
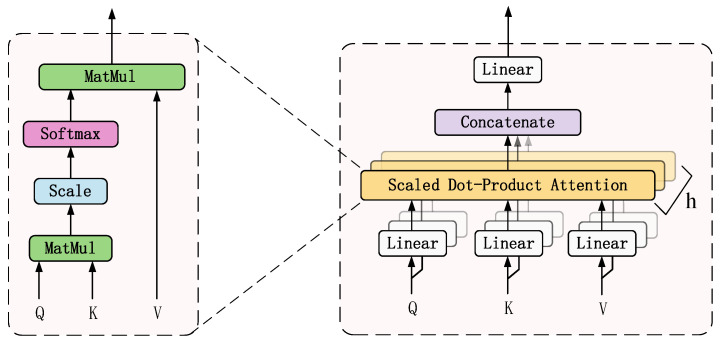
The architecture of multilayer attention layer.

**Figure 7 sensors-22-03878-f007:**
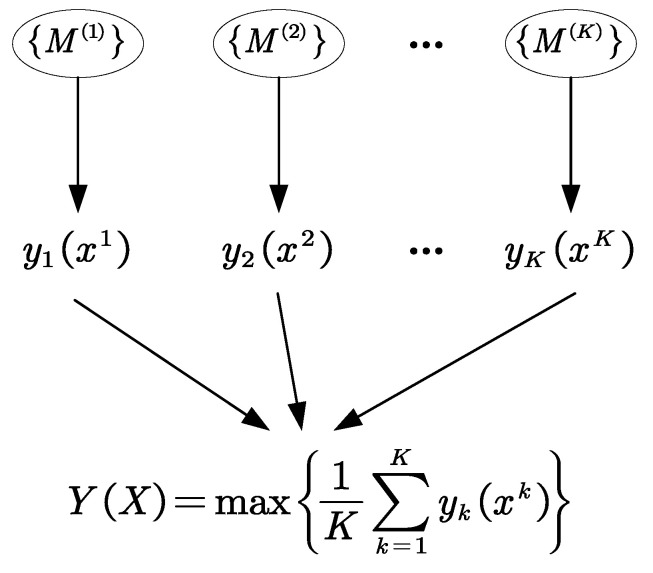
The fusion process of the soft voting method.

**Figure 8 sensors-22-03878-f008:**
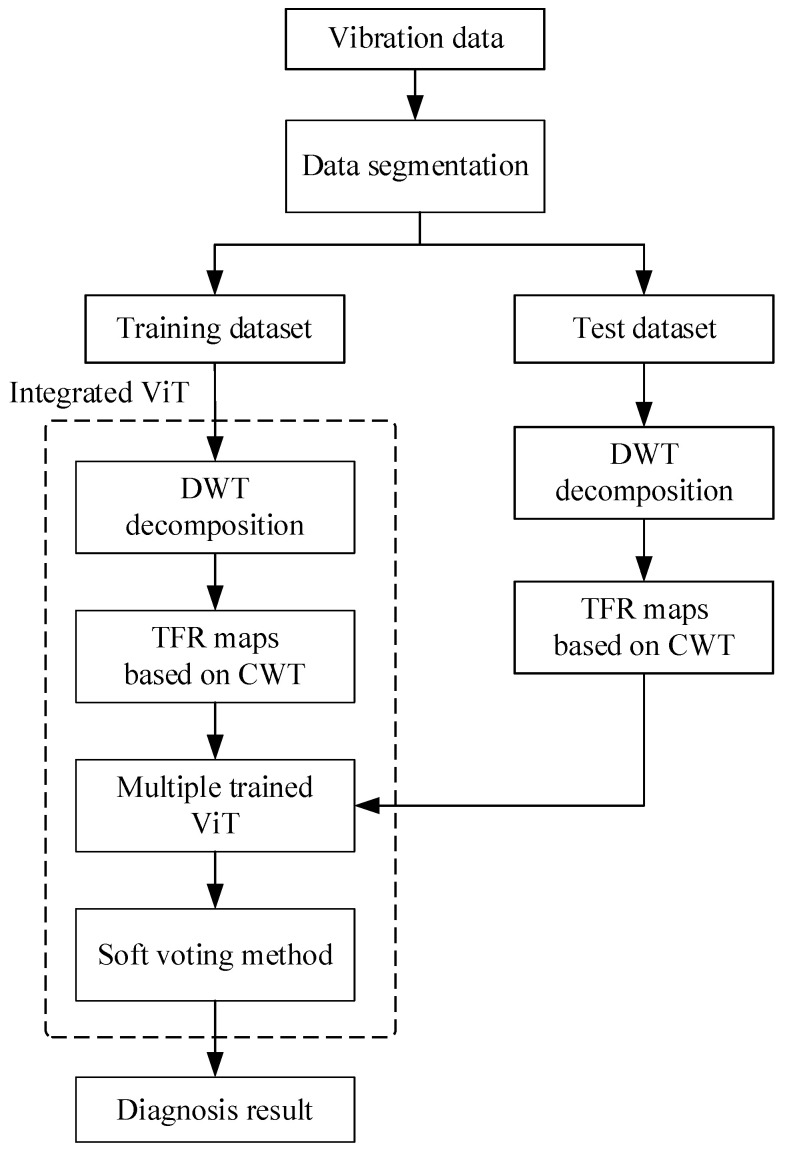
The diagnosis flowchart based on integrated ViT.

**Figure 9 sensors-22-03878-f009:**
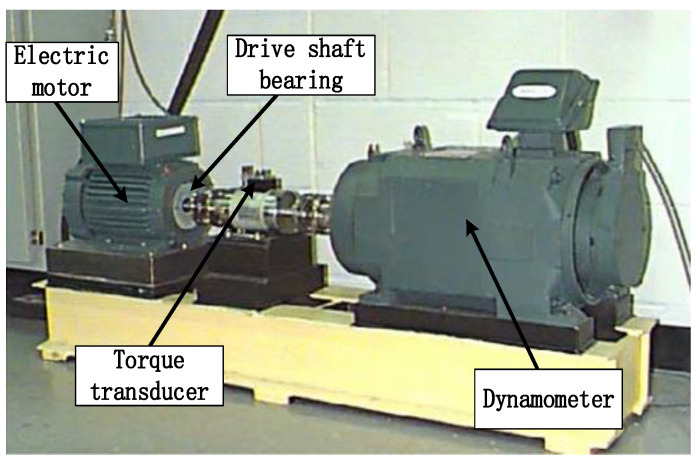
Experimental setup of rolling bearing fault.

**Figure 10 sensors-22-03878-f010:**
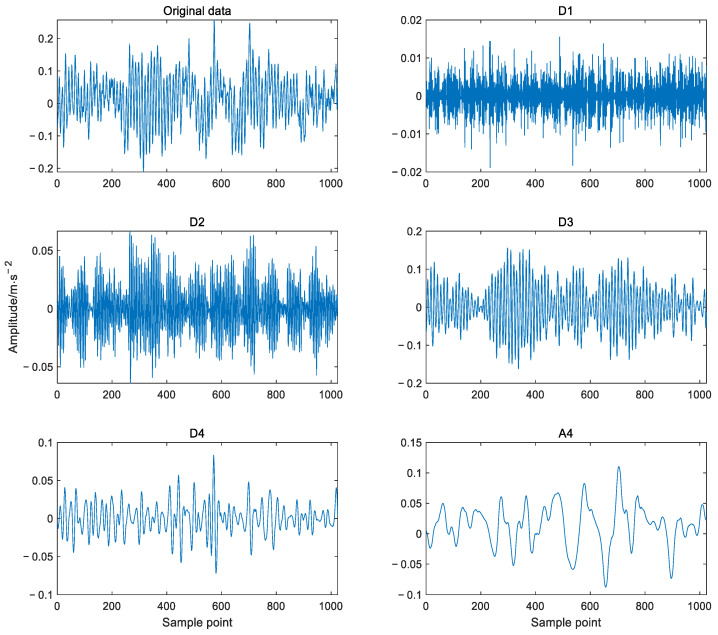
Decomposition of vibration signal based on DWT.

**Figure 11 sensors-22-03878-f011:**
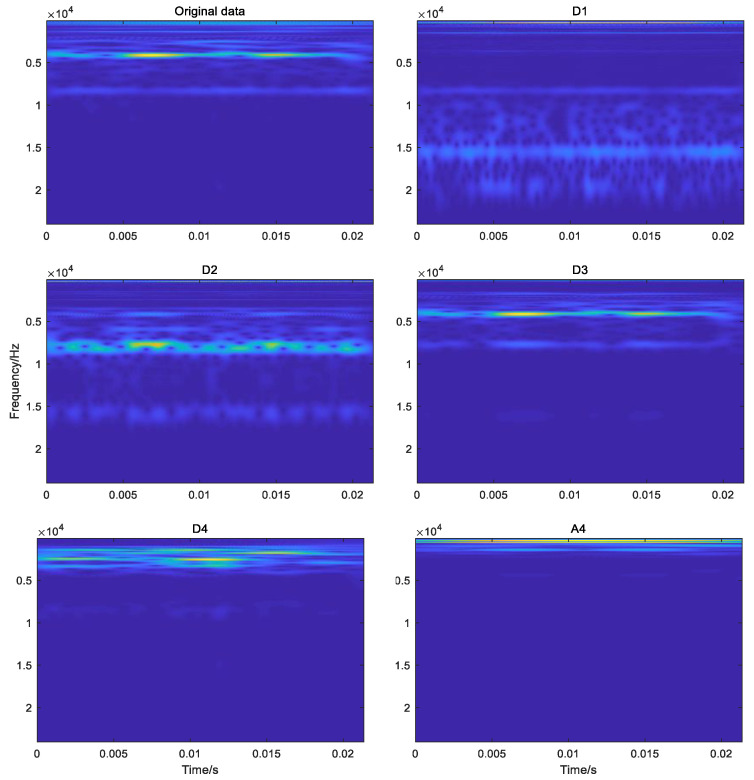
TFR maps of subsignal and original vibration signal.

**Figure 12 sensors-22-03878-f012:**
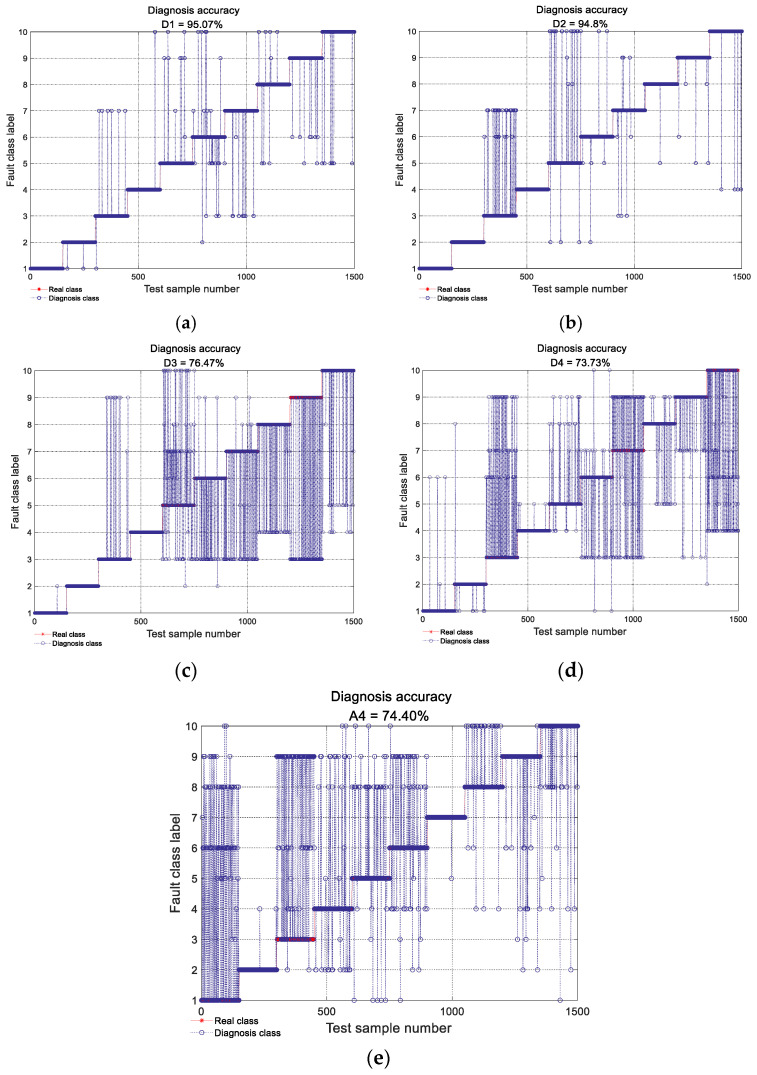
Diagnosis results of an individual ViT with different detail and approximation subsignal TFR. (**a**–**e**) Diagnostic accuracy with an individual ViT with D1, D2, D3, D4 and A4 subsignal TFR, respectively.

**Figure 13 sensors-22-03878-f013:**
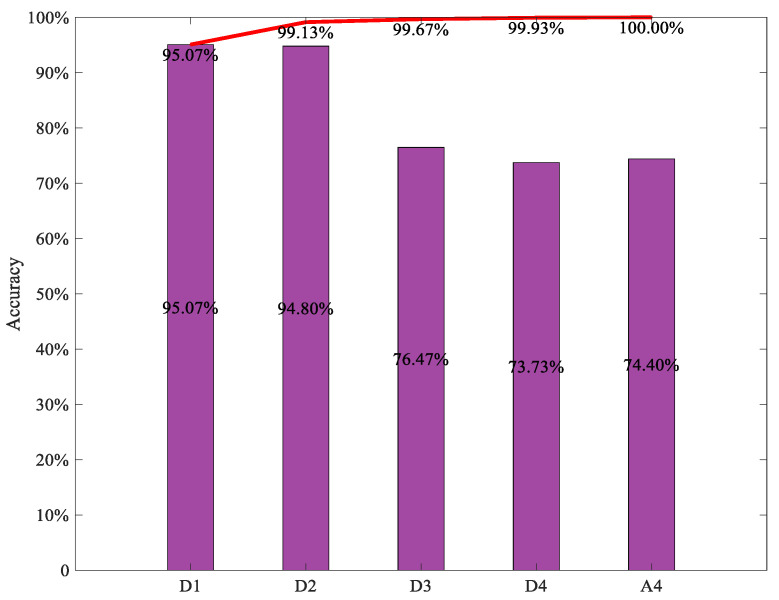
Diagnosis accuracy of the integrated ViT models with first n ViT. The histogram indicates the diagnosis accuracy of the individual ViT with different TFR maps respectively, the red curve indicates the diagnostic accuracy of the integrated ViT model with the first n TFR maps of subsignals.

**Figure 14 sensors-22-03878-f014:**
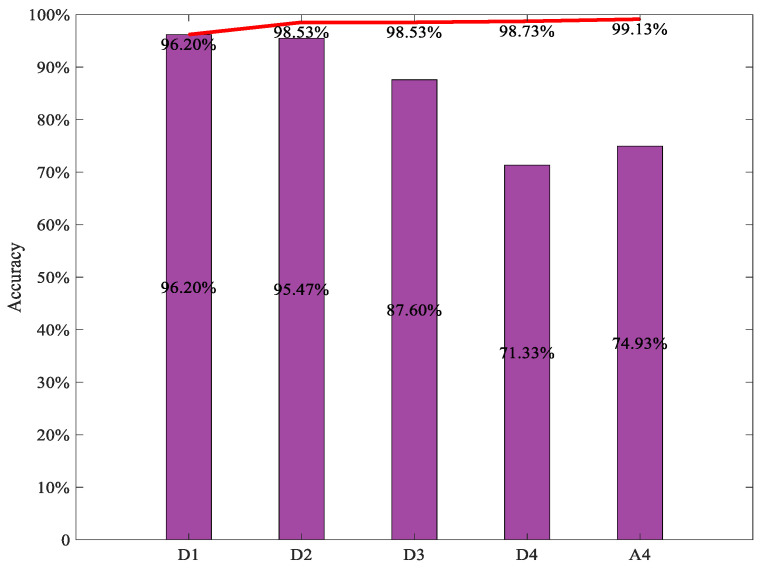
Diagnosis accuracy of integrated CNN using the first n individual CNN. The histogram indicates the diagnosis accuracy of the individual ViT with different TFR maps respectively, the red curve indicates the diagnostic accuracy of the integrated ViT model with the first n TFR maps of subsignals.

**Table 1 sensors-22-03878-t001:** The statistics of bearing fault dataset.

Fault Class Conditions	Class Label	The Number of Training Samples	The Number of Test Samples	Fault Size (mm)
Normal	1	350	150	0
Slight inner ring	2	350	150	0.18
Medium inner ring	3	350	150	0.36
Severe inner ring	4	350	150	0. 53
Slight outer ring	5	350	150	0.18
Medium outer ring	6	350	150	0.36
Severe outer ring	7	350	150	0. 53
Slight rolling element	8	350	150	0.18
Medium rolling element	9	350	150	0.36
Severe rolling element	10	350	150	0.53

**Table 2 sensors-22-03878-t002:** The parameters of CNN model.

Layer	Input Size	Output Size
Conv2D	64, 64, 3	64, 64, 32
Conv2D	64, 64, 32	64, 64, 32
MaxPooling2D	64, 64, 32	32, 32, 32
Flatten	32, 32, 32	32,768
Dense	32,768	32
Dense	32	10

**Table 3 sensors-22-03878-t003:** Performance comparison of individual ViT, integrated ViT and integrated CNN.

Diagnostic Model	Mean of Diagnosis Accuracy	Minimum of Diagnosis Accuracy	Maximum of Diagnosis Accuracy
ViT	98.73%	97.76%	99.87%
Integrated ViT model	99.87%	99.47%	100.00%
Integrated CNN model	99.13%	98.53%	99.87%

**Table 4 sensors-22-03878-t004:** Diagnosis results of three diagnosis models on three datasets.

Diagnosis Model	Diagnosis Accuracy (%)		
	Dataset 1	Dataset 2	Dataset 3
Integrated ViT	100.00	99.67	99.83
Integrated CNN	99.17	99.33	98.33
ViT	98.83	98.67	97.83

## Data Availability

The data used to support this study are available at the website http://csegroups.case.edu/bearingdatacenter/pages/download-data-file (accessed on 24 August 2021).

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
