# Peer review of "A Novel Fault Diagnosis Method of Rolling Bearing Based on Integrated Vision Transformer Model"

_sensors, 2022, doi:10.3390/s22103878_

Round 1

Reviewer 1 Report

In this paper, the Authors are proposing an integrated Vision Transformer (ViT) model based on wavelet transform and soft voting method to improve the diagnosis accuracy and generalization of bearing fault.

The test results showed that the proposed integrated ViT model based on the soft voting method can diagnose the different fault categories and fault severities of bearings accurately. This model has higher accuracy than others models applied to mechanical equipment.

I found that this paper is very interesting and that the obtained results are very promising, however in order to further improve I would only recommend to improve the conclusions and more references on the background (I suggest: doi: 10.3390/su14095335, doi: 10.3390/e22111306, doi: 10.3390/machines10050326, doi: 10.3390/e22111306, doi: 10.3390/e24040511).

Reviewer 2 Report

Dear Authors,

Based on the first round review of the manuscript entitled A novel fault diagnosis method of rolling bearing based on Integrated Vision Transformer model, the reviewer has the following comments:

  1. Please explain about Figure 1 in more details. Please explain all blocks deeply here.
  2.  In Figure 1, if VIT is included sub-blocks, please explain here.
  3. What is the advantages of ViT model compare to other AI or classical signal/system modeling techniques?
  4. How you can validate the stability and robustness of the proposed diagnosis algorithm?
  5. In Figure 8, do the DWT and TFR for training and testing have the same design (coefficients and ....)
  6. Can the proposed method work for acoustic emission (AE) signals as well? (In this manuscript the vibration signal is used to validate the algorithm)
  7. What is the limitation of the proposed method? please mention in the conclusion and based on that please mention about the future work.

Regards,

Reviewer 3 Report

The paper proposes a fault diagnosis approach for the condition monitoring of rolling bearings. The proposed approach uses a combination of Discrete and Continuous Wavelet Transform (DWT and CWT) to decompose the vibrational signal and obtain Time-Frequency representations of the sub-signals. Then, Integrated Vision Transformer (ViT) models are fed with these 2-dimensional, image data in input; soft voting is finally applied to perform damage classification (considering 10 fault classes). This is interesting since damage labelling is quite an advanced task, well above easier aims such as damage detection alone. Hence, the paper has good potential. Another very good point is that the Authors used a well-known case study, the Case Western Reserve University (CWRU) dataset. This allows for both the direct comparison of their results and for the replicability of their algorithm. The manuscript is overall concise, well-written, and easy to follow. The results seem to be good and well-explained from both the figures and the tables in the dedicated section.

Nevertheless, the paper has some critical aspects, both in its content and format, that should be addressed before full acceptance. In detail:

  1. The proposed algorithm is quite complicated and convolute. From this reviewer’s understanding, in step 1, the signal is firstly decomposed into several components by means of Discrete Wavelet Transform (DWT). Then, in step 2, the Continuous Wavelet Transform (CWT) is applied to each of these sub-signals, to obtain Time-Frequency distributions. In step 3, these distributions are then used as damage-sensitive (i.e. fault-related) features for Integrated Vision Transformer (ViT) models. Finally, in the last step, soft voting is applied to fuse all the intermediate results (obtained in step 3) and get a final decision.

This complexity is not necessarily a problem, however, it might arise some doubts, specifically:

  1. The rationale to apply DWT and then CWT is not clear. What are the expected advantages of doing a wavelet transform over a wavelet transform?
  2. Related to the previous remark, if the aim of step 1 is to decompose the original signal into content-related sub-signals/components, there are plenty of data-adaptive algorithms apart from DWT. For instance, in https://doi.org/10.3390/s21051825, the Variational Mode Decomposition (VMD) algorithm was experimentally found to overperform other options (such as CEEMDAN and HVD) for applications to the vibration recordings of mechanical systems. The Authors may consider adding this consideration to their discussion, also discussing why DWT was preferred instead.
  3. A similar consideration can be extended to CWT as well. There are numerous Time–frequency/time-scale (TF/TS) representations, such as the short-time Fourier transform (STFT), the discrete Choi–Williams time-frequency distribution (DCW), the Wigner–Ville distribution (WV), etc that can be used to obtain TFR maps. The motivations behind the selection of CWT should be better highlighted.
  4. Both DWT and CWT rely heavily on the choice of the mother wavelet. In the case reported here, Daubechies of order 5 and cmor3-3 wavelet basis functions have been chosen. However, it is not clear if the Authors did test out other potential options, especially other Db orders. Complex Morlet wavelets as well are not the most usual option; Generalised Morse Wavelet and other alternatives are frequently employed as well. The Authors should discuss their motivations.
  1. Regarding the Vision Transformer algorithm, page 5 (correctly) states “ViT has been applied widely to the field of image and vision recognition because of lower computation power and memory consumption”. To validate this claim, is it possible to have a statistical study on the computational requirements of the proposed methodology? E.g. the elapsed time.
  2. Section 4.3: related to the description made in 2.3.1, the number of pixels along the height h and the width w of the images (the TFR maps) should be indicated.
  3. The method is compared against CNN and individual ViT. However, it is not totally clear if this comparison is enough fair. CNN relies on the optimisation of several hyperparameters. It is not clear if the Authors did perform some fine-tuning to optimise these hyperparameters and obtain the best results possible from their CNN. This aspect should be better discussed.

Other minor (editorial) issues:

  1. While overall well-written, there are a few typos and grammar mistakes in the text. For instance, often the blank space between a word and the following reference is missing (e.g. page 1 line 45 “quickly[10]”.) Similarly, on page 11 line 332, “D1,D2,D3,D4” is written without blank spaces between words. Please double-check carefully.
  2. from an editorial perspective, Fig 5 and 6 are pleasant and well-made. Figure 1, on the other hand, is more basic. Perhaps, it would be better to have the same design for all flowcharts in the manuscript (also for graphical consistency).
  3. Page 3, when referring to the Mallat’s pyramidal algorithm, the original 1989 paper (https://doi.org/10.1109/34.192463) should be referenced.
  4. the font used for the equations is not consistent everywhere (e.g. Eq 1 and 2 have different fonts). Please harmonise it.
  5. Table 3: for consistency, all values should be reported with the same accuracy (that is to say, 100.00 instead of 100).

Round 2

Reviewer 2 Report

Dear Authors,

Regarding the second round review of the manuscript entitled "A novel fault diagnosis method of rolling bearing based on Integrated Vision Transformer model", this manuscript can be accepted for further processing.

Regards,

Author Response

All suggestions have been revised in the paper.

Reviewer 3 Report

In this Reviewer’s opinion, the responses provided by the Authors are not entirely satisfactory. Specifically:

Regarding remark 1.1, it was clear that DWT is applied for signal decomposition and CWT is applied for time-scale (i.e. time-frequency) analysis. The question, therefore, was not about why performing a time-frequency representation of a decomposed signal (the motivation is quite straightforward and related to denoising); the point was about why specifically using wavelet-based approaches for both these tasks. This point has not been addressed in the reply.

Responses 1.2 and 1.3 are not well-argued and mainly only reiterate the (not complete) response to 1.1. As already mentioned in the previous round of remarks, there are several algorithms for signal decomposition and for time-frequency analysis. From the paper and the authors’ reply, there is no evident motivation to prefer the proposed approach to any other candidate.

Response 1.4 is also not very satisfactory. The explanation for the Morlet wavelet is understandable (but still, this information has not been added to the reviewed text of the article). However, the choice of Daubechies of order 5 “mainly based on experience” cannot be considered a valid motivation and seems to imply that the Authors did not test out other Db orders.

The response to point 2 is adequate; potentially, this could have been an interesting point to add to the paper, yet the motivations of the Authors to not include the reported Table 1 are understandable.

The response to point 3 is adequate as well.

Regarding point 4, the authors did not optimise the CNN parameters, not they are intended to, stating that this falls out of the aims of this paper. This is an arguable choice; in any case, they added some more details about the CNN architecture, even if it is not yet complete (e.g. the number of nodes in the layers is not reported).

Regarding the editorial/minor issues, these as well have been only partially addressed. For instance, Figure 1 is now more consistent with the format of the other flowcharts. Nevertheless, the changes are difficult to track since they have not even been replied to in the authors’ reply and some corrections (e.g. “D1, D2, D3, D4” on page 11 and others) have not been highlighted in the revisioned version.

“100%” has been rewritten as “100.00%” but only in Table 2 (not Table 1).

The manuscript still has some typos and grammar errors throughout the whole text (e.g. page 3 line 104“es(Embedding layer),” without a blank space before the bracket).

In conclusion, the authors did not seem to be very receptive to this reviewer’s first round of suggestions; hence, the overall recommendation cannot be changed.
